# Reliability and Validity of the Ground Reaction Force Asymmetric Index at Seat-Off as a Measure of Lower Limb Functional Muscle Strength: A Preliminary Study

Ae-Ryoung Kim [1,†]![], Dougho Park [2,†]![] and Yang-Soo Lee [1,*]![]

1   Department of Rehabilitation Medicine, School of Medicine, Kyungpook National University, Kyungpook National University Hospital, Daegu 41944, Korea; ryoung20@hanmail.net
2   Department of Rehabilitation Medicine, Pohang Stroke and Spine Hospital, Pohang 37659, Korea; parkdougho@gmail.com
*   Correspondence: leeyangsoo@knu.ac.kr; Tel./Fax: +82-53-200-5311
†   Both authors contributed equally to this work.

**Featured Application: The reliability of this newly developed test measuring lower extremity muscle strength using force plate during sit-to stand performance was measured as high, but the validity was poor. To be clinically useful, this test should be further refined by modifying the test protocol and method.**

**Abstract:** This study examined the reliability of the newly developed ground reaction force asymmetry index (GRF AI) at seat-off using a low-cost force plate and the validity of this new test by comparing it with other muscle strength-measuring methods and walking speed. This study was a cross-sectional design in general hospital setting. A convenience sample of 47 community-dwelling adults aged ≥40 years was performed. GRF AI is the measurement value obtained by shifting the weight to the right and left while performing sit-to-stand (STS). GRF AI assessed using GRF data at seat-off during an STS test with maximal weight shift to the right and left side and repeated 4 weeks later. Hip and knee extensor strength were measured using hand-held dynamometry; hand grip strength and walking speed were measured using a standardized method. Intrasessional intrarater reliability of the right and left side at Sessions 1 and 2 were high (intraclass correlation coefficients [ICC] = 0.947 and 0.974; 0.931 and 0.970, respectively). In addition, the intersessional intrarater reliability of a single test trial (ICC = 0.911 and 0.930) and the mean of three test trials (ICC = 0.965 and 0.979) was also high. There was a low correlation between right-side GRF AI and right hand grip strength (r = 0.268) and between left-side GRF AI and left hand grip strength (r = 0.316). No significant correlations were found between the GRF AI and other parameters. Although the reliability of the GRF AI was high, the validity was poor. To be clinically useful, this test should be further refined by modifying the test protocol.

**Keywords:** muscle strength; lower extremity; reliability; validity; force plate

## 1. Introduction

Lower extremity strength is the most important determinant for walking and daily living [1]. Muscle weakness is impaired in diverse medical conditions, including Parkinson's disease [2], motor neuron disease [3], myopathy [4], neuropathy [4], stroke [5], major surgery [6,7], infection [8], and sarcopenia [9]. Impairments in muscle strength can serve as a predictor of important outcomes, such as mortality, hospital length of stay, and hospital readmission [10]. In light of these facts, practical tests for measuring muscle strength are needed. Therefore, a test wherein the examiner objectively and quantitatively measures the loss of strength or muscle power is required. Moreover, for early detection and prevention of muscle weakness progression, it is essential to accurately measure muscle strength. The

manual muscle test (MMT) is a grading scale used to measure muscle strength that was first applied in patients with poliomyelitis in 1915 [11]. While MMT measures muscle strength, the subjective nature of the test lacks reproducibility [10]. In addition to MMT, conventional methods of measuring muscle strength include a hand-held dynamometer (HHD) [12] and isokinetic dynamometry [13]. Evaluations using an HHD must be performed in a specific posture and cannot be performed in all muscles; further, each muscle's strength must be separately measured [14]. It is difficult to assess overall strength by measuring the hand grip strength using a dynamometer. While isokinetic dynamometry is the gold standard for limb strength measurement, it requires a considerable amount of space, an expensive apparatus, and technical expertise. In addition, it can only test specific muscles [15]. In clinical practice, sit-to-stand (STS) and heel-raise tests are widely used to assess muscle strength [10].

These tests [16] typically document the time required to complete a given number of movements such as STS [17] or count the number of repetitions completed in a given time [10]. Despite the frequent use of the STS test, its performance can be affected by balance capability. Therefore, in this study, a new method of measuring functional lower limb muscle strength was developed and tested. The degree to which the weight shifts laterally at seat-off during STS was measured by the ground reaction force (GRF) using a low-cost force plate.

We hypothesize that the GRF asymmetry index (AI) measurement is useful as a method for evaluating lower extremity strength. To verify the validity of the new test, the correlation with the conventional test was analyzed. Therefore, this study aimed to evaluate (1) the reliability of the new GRF AI at seat-off and (2) the validity of this new test by comparing it with other muscle strength measurement methods (HHD and hand grip strength) and walking speed.

## 2. Materials and Methods

### 2.1. Participants

A convenience sample of 47 community-dwelling adults aged ≥40 years participatedin this study. All participants were recruited through personal contact within the hospital. Data were collected from March to August 2020. Written informed consent was obtained from all participants prior to participation. The study protocol was approved by the institutional review board of the hospital (approval no. PSSH-0457-202002-HR-002-01).

Participants with one or more of the following conditions were excluded: (1) musculoskeletal diseases that cause pain during STS; (2) unstable cardiovascular diseases, such as heart failure, acute coronary syndrome, unstable angina, uncontrolled hypertension, and severe left ventricular hypertrophy; (3) previous neurologic disorders or other neurologic conditions that can affect muscle strength and STS; (4) decreased cognition that renders it difficult to perform motions as directed; (5) a history of orthostatic hypotension or syncope; or (6) any other condition that would contraindicate physical effort.

A summary of the baseline characteristics of participants is presented in Table 1. Overall, 47 participants enrolled in the study—26 men and 21 women. The mean age was 59.30 ± 11.26 years (range 40–79), and its distribution was 10 in their 40s, 12 in their 50s, 16 in their 60s, and 9 in their 70s; their ages were almost evenly distributed from 40 to 70 years.

**Table 1.** Demographic characteristics of all participants.

| Characteristics (*n* = 47) | |
| --- | --- |
| Age | 59.30 ± 11.26 |
| 40–49 (*n*) | 10 |
| 50–59 (*n*) | 12 |
| 60–69 (*n*) | 16 |
| 70–79 (*n*) | 9 |
| Male:female (*n*) | 26:21 |
| Height (cm) | 164.09 ± 9.25 |
| Weight (kg) | 63.98 ± 11.73 |
| BMI (kg/m$^2$) | 23.60 ± 11.73 |

Values: mean ± standard deviation. BMI, body mass index.

## 2.2. Procedures

This study consisted of two sessions spaced 4 weeks apart. At the first session, body weight, GRF AI, knee extensor strength, hip extensor strength, grip strength, and walking velocity were measured. At the second session, only GRF AI was measured. The participants maintained their usual daily activities between the two sessions and were instructed not to engage in any special additional physical activity.

All the measurements were collected by two examiners—a physical therapist and an occupational therapist. One educated the participants and performed the evaluation, whereas the other recorded the data. Through training, the examiners were thoroughly familiar with the evaluation and able to correctly implement it. All participants performed each STS movement three times as practice before the actual measurement to familiarize themselves with the test and then underwent the actual examination after sufficient rest. In addition, HHD measurement of the hip and knee and hand grip strength tests were also practiced.

## 2.3. Hand-Held Dynamometry

Isometric lower limb muscle strengths were measured using the microFET2 HHD (Hoggan Health Industries Inc., Salt Lake City, UT, USA) portable HHD, and peak force was expressed in newtons (N). Before the measurement, a 5-kg sand bag was used to calibrate the dynamometer.

Good-to-excellent reliability has been reported for lower limb HHD strength measurement in participants without a neurological condition [18]. Isometric muscle strength of bilateral hip and knee extensors were tested with a standardized test protocol.

To measure knee extensor muscle strength, participant assumed a seated position on the examination table with both palms placed on the table. HHD was applied to the lower third of the participant's tibia and induced the participant to extend the knee with maximum force [19].

To measure hip extensor muscle strength, the participant was instructed to lie in a side lying position on a flat table with the test leg up. Using support materials, the hip joint was positioned at a 0° hip extension and neutral at the coronal plane. The hip and knee of the unmeasured leg were flexed to 90°. The force pad of the HHD was placed on the lower third of the participant's thigh [20]. Once positioned, participants were asked to perform a 3-s maximal contraction. Peak values were recorded for three repetitions. All measurements were obtained by the physical therapist and documented by the occupational therapist. Maximal isometric muscle forces were measured in newtons (N). During the measurements, participants were encouraged to exert maximum effort.

We alternately measured the right and left sides three times with a 1-min break between each contraction, and the mean value of the measurements was used in this analysis. Measured values were normalized to the body weight of each participant.

## 2.4. Hand Grip Strength

The Jamar® Hydraulic Hand Dynamometer (Patterson Medical, Warrenville, IL, USA) was used to measure hand grip strength. Participants assumed a seated position, with their elbow flexed at 90° and the shoulder in a neutral position [21]. They were instructed to maximally squeeze the hand dynamometer for 3 s [21]. The measurement protocol of grip strength was the same as that of HHD.

## 2.5. Walking Speed

We measured each participant's gait speed at a comfortable and self-selected pace. A 10-m walkway was marked out on a flat and smooth floor. Black markers were taped at the 0-, 2-, 8-, and 10-m points of the walkway. Participants were instructed to walk from the 0-m point to the other end of the walkway. The examiner used a stopwatch to measure the time it took a participant to walk from the 2-m to the 8-m mark [22]. Each

participant performed the walk twice, with a 1-min break between trials; the mean of the two measurements was used for analysis.

### 2.6. Ground Reaction Force Asymmetry Index

The GRF AI test measured the difference in the vertical ground reaction force at seat-off between the right and left lower extremities. To measure vertical GRFs, three low-cost force plates (Vernier Software & Technology, Beaverton, OR, USA), about 600 dollars each, were used. The dimensions of the force plates used were 28 × 32 × 5 cm. Therefore, the size of the force plate was sufficient for each foot and both buttocks of the participant. Data were recorded in N and obtained 100 times per second.

For the measurement, participants removed their shoes and placed each foot on the force plate of the corresponding side while sitting on a force plate on the motorized table, adjusted according to each participant's height. Before STS, the participant spread both feet to shoulder width. Table height was adjusted by examiner such that the participant's ankle joint dorsiflexion was at 15°, the knee joint flexion was at 105°, and the thigh was parallel to the ground. The examiner ensured that the participant's trunk was vertical and that the table only touched approximately one third of the participant's thigh. Because the participant was sitting on the force plate, weight bearing on the Bobath table was minimal while their hands were touching the wall.

For the GRF AI of the maximal weight shift to the right, participants slightly touched the wall with their hands at chest level and then stood upright while maximally moving weight to the right. The procedure was identical for the GRF AI of the maximal weight shift to the left, except participants maximally shifted their weight to the left while standing. Participants were instructed to keep both feet stable on the force plates during the test to enhance safety. To prevent sudden movement, a metronome was used during the test and participants were instructed to smoothly perform each action for 2 s. The GRF data of the lower extremities were obtained when the GRF of the buttocks reached zero, that is, when the buttocks were detached from the force plate (seat-off) (Figure 1). Hip and knee extension torques reach their peak values around the instant of seat-off [23].

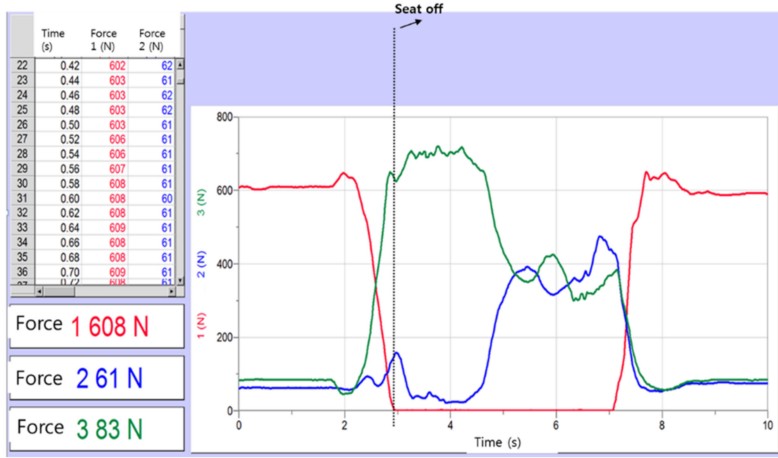

**Figure 1.** Force plate graph of left-side weight shifting during the sit-to-stand performance. Red line, buttock force plate; blue line, right foot force plate; green line, left foot force plate.

Participants performed nine STS performances, three times in each STS method (three with even weight distribution, three with maximal weight shift to the right side, and three with maximal weight shift to the left side). Each standing method was alternately performed. A 1-min rest was given between trials to prevent fatigue.

GRF AI is the value obtained by shifting the weight to the right and left performing STS. The GRF AI was defined as the GRF value of each side divided by the sum of both sides. The equation of the GRF AI was as follows:

GRF AI at maximal weight shift to right (GRF AI Rt) = right GRF/(right GRF + left GRF)

GRF AI at maximal weight shift to left (GRF AI Lt) = left GRF/(right GRF + left GRF).

### 2.7. Statistical Analysis

SPSS version 23.0 (IBM Inc., Armonk, NY, USA) [24] was used for all data analyses. Descriptive statistics were calculated. Descriptive data for demographic characteristics, lower extremity muscle strength, grip strength, walking velocity, and GRF AI were recorded. All tests were conducted by same rater, so only intrarater reliability was evaluated.

Assessment of the intrarater reliability of GRF ratio was conducted using intraclass correlation coefficients (ICCs) with 95% confidence intervals. A two-way mixed model with absolute agreement was used for the calculation. ICCs were used to evaluate intersessional and intrasessional reliability due to the single rater design. Interpretations of ICC values were made according to the following scale: poor (<0.69), fair (0.70–0.79), good (0.80–0.89), and high (0.90–1.00) [25]. Intersessional reliability was analyzed in two ways. The first method (intersessional test–retest reliability using the mean of the three repeated measurements) involved the comparison of the mean values of the three trials on each session, and the second method (intersessional test–retest reliability using single test) involved the comparison of the first trial of each session. Paired sample t test was used to compare the GRF AI of the first and second sessions.

Concurrent validity was assessed using the Pearson correlation to compare the GRF AI with HHD measurement, hand grip strength, and walking speed. Based on a previous study [23] that reported a correlation between the peak torque of the hip and knee extensors and GRF values during the STS, the relationship between the total value of hip and knee measured by HHD and GRF AI at seat-off was analyzed. Pearson correlation coefficients were defined as high (>0.70), moderate (0.50–0.69), low (0.26–0.49), and little-to-no correlation (0.00–0.25) [26]. $p < 0.05$ was considered statistically significant.

## 3. Results

### 3.1. Participant Characteristics

Mean values of the GRF AI of the 47 participants collected from the two different sessions are presented in Table 2. Both right and left GRF AI values were between 0.68 and 0.69. In addition, representative data of the HHD measurement, hand grip strength, and walking speed are presented in Table 2. There were no significant differences in measurements between the two test trials.

**Table 2.** Ground reaction force asymmetry index, hand-held dynamometer, grip strength, and walking speed.

| | Day 1 | | Day 2 | | Mean Diff. between Trial | |
|---|---|---|---|---|---|---|
| | **Right Side** | **Left Side** | **Right Side** | **Left Side** | **Right Side** | **Left Side** |
| GRF AI (ratio) | | | | | | |
| Neutral STS | 0.488 ± 0.038 | 0.512 ± 0.038 | 0.487 ± 0.044 | 0.513 ± 0.044 | 0.002 ± 0.032 | 0.001 ± 0.032 |
| Each side weight-bearing STS | 0.685 ± 0.094 | 0.685 ± 0.083 | 0.689 ± 0.073 | 0.695 ± 0.062 | 0.004 ± 0.041 | 0.014 ± 0.039 |
| Hand-held dynamometer (N) | | | | | | |
| Hip extensor | 0.216 ± 0.072 | 0.210 ± 0.066 | | | | |
| Knee extensor | 0.413 ± 0.075 | 0.408 ± 0.073 | | | | |
| Grip strength (lbs) | 0.106 ± 0.022 | 0.100 ± 0.024 | | | | |
| Walking speed (m/s) | 4.490 ± 0.573 | | | | | |

Values: mean ± standard deviation. STS, sit-to-stand; GRF AI, ground reaction force asymmetry index; N, newtons; m/s, meter/second.

### 3.2. Reliability

The intrasessional intrarater reliability of the GRF AI is presented in Table 3. The intrasessional intrarater reliability of both sessions was high: ICCs of the right and left side in Session 1 were 0.947 and 0.974, respectively, whereas those in Session 2 were 0.931 and 0.970, respectively (Table 3a).

**Table 3.** Reliability of sit-to-stand test performance conducted two times 1 month apart (*n* = 47).

| | (a) Intrarater Reliability of Mean GRF Ratio | | | | | |
|---|---|---|---|---|---|---|
| | Intrarater ICC (CI)_day 1 | | Intrarater ICC (CI)_day 2 | | Intrarater ICC (CI) | |
| | Right side | Left side | Right side | Left side | Right side | Left side |
| GRF AI | 0.947 (0.914–0.969) | 0.974 (0.959–0.985) | 0.931 (0.889–0.959) | 0.970 (0.952–0.982) | 0.965 (0.948–0.979) | 0.979 (0.969–0.987) |
| | (b) Single-Time Test–Retest Reliability | | | | | |
| | Intrarater ICC (CI) | | | | | |
| | Right side | | | Left side | | |
| GRF AI | 0.911 (0.815–0.947) | | | 0.930 (0.874–0.961) | | |

GRF AI, ground reaction force asymmetry index; ICC, intraclass correlation coefficient; CI, confidence interval

Further, the intersessional test–retest reliability using the mean of three repeated measurements (4-week interval) was high (ICC, 0.965 and 0.979) (Table 3a). The intersessional test–retest reliability of the GRF AI using a single test was high between the first measurement value at Session 1 and the first measurement value at Session 2 (0.911 and 0.930) (Table 3b).

### 3.3. Validity

The relationship between GRF AI, knee extensor strength, hip extensor strength, hand grip strength, and walking speed is presented in Table 4.

**Table 4.** Concurrent validity analysis of force plate (r, Pearson correlation) (*n* = 47).

| | Rt Hip Knee Extensor | Rt Hand Grip | Walking Speed |
|---|---|---|---|
| GRF AI right neutral STS | 0.034 (0.072) | 0.191 (0.251) | 0.165 (0.488) |
| GRF AI Rt | 0.045 (0.359) | 0.268 * (0.042) | 0.187 (0.115) |
| | **Lt Hip Knee Extensor** | **Lt Hand Grip** | **Walking Speed** |
| GRF AI left neutral STS | 0.094 (0.591) | 0.217 (0.238) | 0.044 (0.278) |
| GRF AI Lt | 0.064 (0.654) | 0.316 * (0.009) | 0.117 (0.437) |

* Correlation is significant at the 0.05 level; ( ), *p* value. GRF AI, ground reaction force asymmetry index; Rt, right; Lt, Left; GRF AI Rt, ground reaction force asymmetry index at maximal weight shift to the right at seat-off; GRF AI Lt, ground reaction force asymmetry index at maximal weight shift to the left at seat-off.

Pearson correlation coefficients indicated a low correlation between GRF AI Rt and right hand grip strength (r = 0.268) and between GRF AI Lt and left hand grip strength (r = 0.316). No significant correlations were observed between GRF AI and the other outcome measures (Table 4). HHD and hand grip strength showed moderate-to-high correlation (r = 0.896–0.913) in both the right and left sides (Table 5). A low correlation was found between right and left hand grip strength and walking speed, with Pearson correlation coefficients of 0.375 and 0.302, respectively (Table 5).

For additional validity analysis, we attempted to analyze outliers using the 1.5 (IQR) rule [27]. Therefore, the statistics of GRF AI Rt and GRF AI Lt were analyzed, but the outlier part was not confirmed.

**Table 5.** Correlation among muscle strength analysis (r, Pearson correlation) (*n* = 47).

| | Rt Hip Knee Extensor | Lt Hip Knee Extensor | Rt Hand Grip | Lt Hand Grip | Walking Speed |
|---|---|---|---|---|---|
| Rt hip knee extensor | - | - | - | - | - |
| Lt hip knee extensor | 0.896 † (<0.001) | - | - | - | - |
| Rt hand grip | 0.325 † (<0.001) | 0.255 (0.084) | - | - | - |
| Lt hand grip | 0.333 † (<0.001) | 0.323 † (<0.001) | 0.913 † (<0.001) | - | - |
| Walking speed | 0.028 (0.850) | 0.079 (0.600) | 0.375 * (0.009) | 0.302 * (0.039) | - |

\* Correlation is significant at the 0.05 level. † Correlation is significant at the 0.01 level. Rt, right; Lt, Left.

## 4. Discussion

To the best of our knowledge, this is the first study to investigate the reliability and validity of GRF AI in community-dwelling adults at seat-off using a low-cost force plate. The results of this study indicate that the GRF AI has high intrasession and intersession intrarater reliability and poor validity for lower limb muscle strength. The GRF AI is a newly developed method of measuring functional lower limb muscle strength using STS and a low-cost force plate. In contrast to other tests using STS, the GRF AI could measure unilateral lower limb muscle strength and is less vulnerable to balance problems. Before our study, Zemková et al. [28] performed a study using a force plate during chair-rising in active older adults. They revealed that peak power and velocity during chair-rising were reliable and valid indicators of lower body power. Meanwhile, our study was significant in that it provided a novel measurement method and results by defining GRF AI. It is expected that the GRF AI enables the ready measurement of the lower extremity muscle strength for not only the general population but also the patients who have asymmetric lower extremity weakness.

High reliability between test–retests is the most important factor in new test methods [29]. In this study, the GRF AI showed high reliability within the measured values of each test performed three times on the same session and between test session 1 and 2, with no significant difference in the results. Moreover, the single-time test–retest reliability was confirmed as high, indicating that even one test result is useful. This indicates that the test protocol is sufficiently familiarized for the test. Confirming the high reliability of this test demonstrates the possibility that it can be used as a diagnostic or patient evaluation method.

To date, there has been no objective evaluation scale of lower extremity muscle strength that can be used in the clinical setting; therefore, muscle strength or functional status was indirectly predicted by grip strength [30]. STS performance is a basic motion required in daily life that is easy for the participant to perform, and it can be used to directly measure lower extremity muscle strength [12,31]. The 30-s STS test (30STS) records the maximum number of STS movements a participant can perform in 30 s [32]. It can be used to assess exercise capacity, and previous studies have revealed that 30STS appears suitable to evaluate exercise tolerance. However, the participant can easily experience fatigue after 30STS [33]. The five-repetition STS test (FRSTST) measures the time taken to complete five repetitions of STS movement and is used as an indicator of lower extremity strength [34]. FRSTST can be easily applied in various settings and provides an objective measurement of lower extremity muscle strength; however, age, body weight, and stature influence FRSTST and should be considered [35]. Furthermore, some participants face difficulty completing the evaluation, which is a limitation [36]. Previous studies on test–retest reliability reported that grip strength has an ICC of 0.91~0.95 [37], while that of 30STS is ICC 0.87 [38], and FRSTS is ICC 0.64~0.96 [34]. Even when compared with these results, high reliability is confirmed by test–retest of the GRF AI.

There are several advantages of the GRF AI compared with the conventional STS test. First, it is less susceptible to balance-related issues because of the inclusion of the wall touch. Bohannon recommend that participants rest hands on the wall to reduce instability [10]. Second, it is less susceptible to endurance issues of participants because

it required only one to three STS. Third, it can assess the right and left lower extremity muscle strength individually.

There are numerous studies that use force plates to measure muscle strength. In previous studies, two force plates were used to evaluate the weight bearing of the right or left side of the foot, without touching the wall with the hand, and the measured values were peak ground force and the rate of force development [39,40]. Performing the STS test without wall touch reflects lower extremity muscle strength and balance ability [41]. We believe that it is possible to measure pure muscle strength by performing STS with the hand slightly touching the wall, and it improves safety when performing the movement.

The relationship between the GRF during STS and lower extremity strength remains unclear. Yamada and Demura reported that in older women, GRF showed a moderate correlation with isometric knee extension muscle strength [42]. Tsuji et al. revealed that the GRF parameter in STS is related to muscle strength in the knee and ankle [40]. However, this measurement method is not yet practically used as a field test, owing to the requirement of expensive special equipment and the lack of an accurate execution protocol for chair height and time when performing STS [40]. On the other hand, our study has an advantage in that it provides a method for measuring lower extremity muscle strength using GRF with a relatively simple and applicable setting.

No significant correlation was observed between the hip and knee strength measured by HHD and GRF AI, considering that the Pearson correlation coefficient was approximately 0.034–0.094. The new method measured closed kinetic chain muscle strength; however, the use of HHD for open kinetic chain muscle strength may explain this result [43,44]. Therefore, we planned to test closed kinetics strength using the inverse dynamics method [45].

In addition, no significant correlation was observed between the GRF AI and hand grip strength. This could be explained by the different muscles tested. Grip strength measures the strength of the forearm muscles, whereas the GRF AI measures the strength of lower limb muscles.

Another explanation is the limitation of the new test—the ceiling effect. GRF AI was distributed at approximately 0.69. If the ceiling effect could be reduced, the correlation between the two tests may be improved.

HHD and grip strength showed a high correlation on the right and left sides, respectively, with a Pearson correlation coefficient of 0.896–0.913. This finding was consistent with the correlation between grip strength and muscle strength measured by HHD in previous studies [18,30].

There was no significant correlation between the new test and walking speed. Because the study was conducted in relatively healthy participants aged ≥40 years, it was considered that the correlation between walking speed and other muscle strength evaluations was not high, as the minimum strength required for performing tasks is different and reserve capacity exists in healthy individuals [46]. It may be the ceiling effect of this new test for measuring lower extremity muscle strength. Thus, the chair heights were lowered during the STS performance to make the performer's muscular strength more necessary [47]. The change in seat height may decrease the ceiling effect of the new test. Comparison between different seat heights could be facilitated using biomechanics equation.

The most likely explanation for the lack of correlation in this study might be originated from the protocol. There may be insufficient time to shift weight maximally during STS. The study protocol should be changed to improve validity. In the future, the protocol modification for the precise measurements should be considered. In particular, while performing STS, the modified protocol should include that the subject's weight can be sufficiently shifted to the right or left side. In addition, the height of a chair should be appropriately adjusted in the modified protocol.

There are some limitations in this study. First, validity analysis was not compared with the isokinetic dynamometer—the gold standard for measuring muscle strength. Second, the study results did not show the validity of the new test. A comparison with closed

kinetic muscle strength using inverse dynamics is necessary to confirm concurrent validity. To improve criterion validity, modification of the protocol may be required. A change in seat height is one possible solution. Third, the method of weight shifting when performing the STS movement may differ among study participants. Although a detailed method was suggested, the STS test is greatly affected by the chair height and feet position [48]. Therefore, we intend to change the protocol in a subsequent study to ensure that more accurate measurements are possible.

## 5. Conclusions

The reliability of GRF AI, which is a newly developed test, was determined to be high (ICC = 0.947, 0.974), although its validity was relatively poor (r = 0.268, 0.316) with hand grip strength. For use as an effective clinical test, this test should be further refined by modifying the test protocol.

**Author Contributions:** Conceptualization, A.-R.K., D.P. and Y.-S.L.; methodology, A.-R.K., D.P. and Y.-S.L.; formal analysis, A.-R.K., D.P. and Y.-S.L.; investigation, A.-R.K., D.P. and Y.-S.L.; writing—original draft preparation, A.-R.K., D.P. and Y.-S.L.; writing—review and editing, A.-R.K., D.P. and Y.-S.L. All authors have read and agreed to the published version of the manuscript.

**Funding:** This research received no external funding.

**Institutional Review Board Statement:** The study was conducted according to the guidelines of the Declaration of Helsinki and approved by the Institutional Review Board of Pohang Stroke and Spine Hospital (approval no. PSSH-0457-202002-HR-002-01).

**Informed Consent Statement:** Informed consent was obtained from all subjects involved in the study.

**Data Availability Statement:** The data presented in this study are available on request from the corresponding author.

**Acknowledgments:** The authors are grateful to Young Hwan Jang and Min Sol Kim for their technical support and dedication to this project.

**Conflicts of Interest:** The authors declare no conflict of interest.

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
