# Peer review of "Reliability and Validity of the Ground Reaction Force Asymmetric Index at Seat-Off as a Measure of Lower Limb Functional Muscle Strength: A Preliminary Study"

_applsci, doi:10.3390/app11146527_

Round 1

Reviewer 1 Report

The study examines the reliability of the newly developed ground reaction force asymmetry index (GRF AI) at seat-off using a low-cost force plate and the validity of this new test by comparing it with other muscle strength-measuring methods and walking speed. The authors concluded that “the reliability of this new developed test measuring lower extremity muscle strength using force plate during sit-to stand performance was measured as high, but the validity was poor”.

This study adds novel but not significant data to the existing knowledge. There are major concerns with respect to the argumentation in favour of this research.

Please, define the ground reaction force asymmetry index (GRF AI).

Page 1, lines 22-24: GRF AI was assessed using GRF data at seat-off during a sit-to-stand test with maximal weight shift to the right and left side and repeated 4 weeks later.

The introduction does not concisely focus on those aspects that would argue in favor of the necessity and novelty of this research. It should be considerably restructured. More information on a variety of sit-to-stand tests should be included.

In addition, the rationale for investigating the relationship between ground reaction force asymmetry index and other functional variables (e.g., hand grip strength) should be introduced.

Page 1, lines 29-31: There was a low correlation between right-side GRF AI and right hand grip strength (r = 0.268) and between left-side GRF AI and left hand grip strength (r = 0.316). No significant correlations were found between the GRF AI and other parameters.

Please, set up the hypothesis of the study.

Page 2, lines 66-68: This study aimed to evaluate (1) the reliability of the new GRF asymmetry index (AI) at seat-off and (2) the validity of this new test by comparing it with other muscle strength measurement methods and walking speed.

Specify the age range.

Page 2, lines 71-72: A convenience sample of 47 community-dwelling adults aged ≥40 years participated in this study.

Please, include information on participant's physical and other activities during four weeks between test sessions which could influence results of the study.

Page 2, line 84: This study consisted of two sessions spaced 4 weeks apart.

Please, consider to remove Figure 1.

Page 4, lines 171-172: Figure 1. Measurement of ground reaction force asymmetry index at seat-off during the sit-to-stand performance.

Remove information on “Participant characteristics” from the Results section and included it in a chapter 2.1. “Participants”.

Page 5, lines 202-205:

3.1. Participant characteristics

A summary of the baseline characteristics of participants is presented in Table 1. Overall, 47 participants enrolled in the study—26 men and 21 women. The mean age was 59.30 ± 11.26 years, and their ages were almost evenly distributed from 40 to 70 years.

Please, include measures for each variable used.

Table 2. Ground reaction force asymmetry index, hand-held dynamometer, grip strength, and walking speed.

Consider to remove Figures 3A and 3B.

Page 7: Figure 3. Correlation of ground reaction force (GRF) asymmetry index (AI) on days 1 and 2

Consider to remove Figures 4A and 4B.

Page 9: Figure 4. Correlation of ground reaction force (GRF) asymmetry index (AI) with hip and knee extensor muscle strength

Consider to remove Figures 5A and 5B.

Page 10: Figure 5. Exploratory ground reaction force (GRF) asymmetry index (AI) analysis

To the best of our knowledge, this is the first study to investigate the reliability and validity of GRF AI in community-dwelling adults.

See also: Zemková E, Jeleň M, Schickhofer P, Hamar D: Jumping From a Chair is a More Sensitive Measure of Power Performance In Older Adults Than Chair Rising. Exp Aging Res 2016;42(5):418-430. doi: 10.1080/0361073X.2016.1224665.

The discussion needs to present practical applications of findings with respect to community-dwelling adults that are not currently addressed in the literature. 

Specify “newly developed test” and include main findings on its reliability and validity.

Page 12, lines 444-446: The reliability of this newly developed test was determined to be high, although its validity was poor. For use as an effective clinical test, this test should be further refined by modifying the test protocol.

Author Response

Response to Reviewer 1 Comments

We sincerely thank the reviewers for their constructive and valuable comments. We have thoroughly addressed their specific concerns and have introduced the suggested changes that were very helpful in improving the manuscript. The main changes in the revised manuscript are the red font color. We used the “Track Changes” function in Microsoft Word. Point-by-point responses to the reviewers’ comments are provided below.

The study examines the reliability of the newly developed ground reaction force asymmetry index (GRF AI) at seat-off using a low-cost force plate and the validity of this new test by comparing it with other muscle strength-measuring methods and walking speed. The authors concluded that “the reliability of this new developed test measuring lower extremity muscle strength using force plate during sit-to stand performance was measured as high, but the validity was poor”.

This study adds novel but not significant data to the existing knowledge. There are major concerns with respect to the argumentation in favour of this research.

Please, define the ground reaction force asymmetry index (GRF AI).

Page 1, lines 22-24: GRF AI was assessed using GRF data at seat-off during a sit-to-stand test with maximal weight shift to the right and left side and repeated 4 weeks later.

  • We define the ground reaction force asymmetry index (GRF AI) as followed. Line 22-23 : GRF AI is the measurement value obtained by shifting the weight to the right and left while performing sit to stand (STS).
  • Line 175-176 : GRF AI is the value obtained by shifting the weight to the right and leftwhile performing STS. The GRF AI was defined as the GRF value of each side divided by the sum of both sides. The equation of the GRF AI was as follows: · GRF AI at maximal weight shift to right (GRF AI Rt) = right GRF/(right GRF + left GRF) · GRF AI at maximal weight shift to left (GRF AI Lt) = left GRF/(right GRF + left GRF).

The introduction does not concisely focus on those aspects that would argue in favor of the necessity and novelty of this research. It should be considerably restructured. More information on a variety of sit-to-stand tests should be included.

  • We modified as followed.
  • Line 61-62 : These tests typically document the time required to complete a given number of movements, such as STS, or count the number of repetitions completed in a given time.
  • Line 68-72 : We hypothesize that the GRF asymmetry index (AI) measurement is useful as a method for evaluating lower extremity strength. To verify the validity of the new test, the correlation with the conventional test was analyzed. Therefore, this study aimed to evaluate (1) the reliability of the new GRF AI at seat-off and (2) the validity of this new test by comparing it with other muscle strength measurement methods (HHD and hand grip strength) and walking speed.

In addition, the rationale for investigating the relationship between ground reaction force asymmetry index and other functional variables (e.g., hand grip strength) should be introduced.

Page 1, lines 29-31: There was a low correlation between right-side GRF AI and right hand grip strength (r = 0.268) and between left-side GRF AI and left hand grip strength (r = 0.316). No significant correlations were found between the GRF AI and other parameters.

  • We modified as followed.

Line 69-70 : To verify the validity of the new test, the correlation with the conventional test was analyzed.

Please, set up the hypothesis of the study.

Page 2, lines 66-68: This study aimed to evaluate (1) the reliability of the new GRF asymmetry index (AI) at seat-off and (2) the validity of this new test by comparing it with other muscle strength measurement methods and walking speed.

  • We have been modified and added as follows.
  • Line 67-69 : We hypothesize that the GRF asymmetry index (AI) measurement is useful as a method for evaluating lower extremity strength.

Specify the age range.

Page 2, lines 71-72: A convenience sample of 47 community-dwelling adults aged ≥40 years participated in this study.

  • Line 90-92 : The age distribution of the participants was 10 in their 40s, 12 in their 50s, 16 in their 60s, and 9 in their 70s.

Please, include information on participant's physical and other activities during four weeks between test sessions which could influence results of the study.

Page 2, line 84: This study consisted of two sessions spaced 4 weeks apart.

  • Line 96-98 : The participants maintained their usual daily activities between the two sessions and were instructed not to engage in any special additional physical activity.

Please, consider to remove Figure 1.

Page 4, lines 171-172: Figure 1. Measurement of ground reaction force asymmetry index at seat-off during the sit-to-stand performance.

  • We removed Figure 1 in response to review comments.

Remove information on “Participant characteristics” from the Results section and included it in a chapter 2.1. “Participants”.

Page 5, lines 202-205:

3.1. Participant characteristics

A summary of the baseline characteristics of participants is presented in Table 1. Overall, 47 participants enrolled in the study—26 men and 21 women. The mean age was 59.30 ± 11.26 years, and their ages were almost evenly distributed from 40 to 70 years.

We remove information 3.1. Participant characteristics and included it in 2.1. Participants.
Line 88-89: A summary of the baseline characteristics of participants is presented in Table 1. Overall, 47 participants enrolled in the study—26 men and 21 women. The mean age was 59.30 ± 11.26 years (range 40–79), and its distribution was 10 in their 40s, 12 in their 50s, 16 in their 60s, and 9 in their 70s; their ages were almost evenly distributed from 40 to 70 years.

Please, include measures for each variable used.

Table 2. Ground reaction force asymmetry index, hand-held dynamometer, grip strength, and walking speed.

  • We include measurement unit for each variable such as GRF AI, hand-held dynamometer (hip extensor, knee extensor), grip strength, walking speed in Table 2.

Consider to remove Figures 3A and 3B.

Page 7: Figure 3. Correlation of ground reaction force (GRF) asymmetry index (AI) on days 1 and 2

  • We removed Figure 3 in response to review comments.

Consider to remove Figures 4A and 4B.

Page 9: Figure 4. Correlation of ground reaction force (GRF) asymmetry index (AI) with hip and knee extensor muscle strength

  • We removed Figure 4 in response to review comments

Consider to remove Figures 5A and 5B.

Page 10: Figure 5. Exploratory ground reaction force (GRF) asymmetry index (AI) analysis

  • We removed Figure 5 in response to review comments.

To the best of our knowledge, this is the first study to investigate the reliability and validity of GRF AI in community-dwelling adults.

See also: Zemková E, Jeleň M, Schickhofer P, Hamar D: Jumping From a Chair is a More Sensitive Measure of Power Performance In Older Adults Than Chair Rising. Exp Aging Res 2016;42(5):418-430. doi: 10.1080/0361073X.2016.1224665.

  • We modified as followed.
  • Line 366-373 : Before our study, Zemková et al. performed a study using a force plate during chair-rising in active older adults. They revealed that peak power and velocity during chair-rising were reliable and valid indicators of lower body power. Meanwhile, our study was significant in that it provided a novel measurement method and results by defining GRF AI. To the best of our knowledge, this is the first study to investigate the reliability and validity of GRF AI in community-dwelling adults at seat-off using a low-cost force plate.

The discussion needs to present practical applications of findings with respect to community-dwelling adults that are not currently addressed in the literature.

  • We have added practical application content to the discussion section.

Line 370-373 : It is expected that the GRF AI enables readily measure the lower extremity muscle strength for not only the general population but also the patients who have asymmetric lower extremity weakness.

Specify “newly developed test” and include main findings on its reliability and validity.

Page 12, lines 444-446: The reliability of this newly developed test was determined to be high, although its validity was poor. For use as an effective clinical test, this test should be further refined by modifying the test protocol.

  • We modified as followed.
  • Line 458-460 : The reliability of GRF AI, which is a newly developed test, was determined to be high (ICC = 0.947, 0.974), although its validity was relatively poor (r=0.268, 0.316) with hand grip strength.

Reviewer 2 Report

Dear authors, 

I have carefully read your manuscript. I think it is interesting, even though the results have not been optimal. 

I consider that you have to include a series of recommendations on how to improve the equipment, since you, through your study, have a broad knowledge of the way forward. Please include a paragraph about this in your conclusions.

Author Response

Response to Reviewer 2 Comments

We sincerely thank the reviewers for their constructive and valuable comments. We have thoroughly addressed their specific concerns and have introduced the suggested changes that were very helpful in improving the manuscript. The main changes in the revised manuscript are the red font color. We used the “Track Changes” function in Microsoft Word. Point-by-point responses to the reviewers’ comments are provided below.

Dear authors,

I have carefully read your manuscript. I think it is interesting, even though the results have not been optimal.

I consider that you have to include a series of recommendations on how to improve the equipment, since you, through your study, have a broad knowledge of the way forward. Please include a paragraph about this in your conclusions.

  • We modified as followed.

Line 454-458 : In the future, the protocol modification for the precise measurements should be considered. In particular, while performing STS, the modified protocol should include that the subject's weight can be sufficiently shifted to the right or left side. In addition, the height of a chair should be appropriately adjusted in the modified protocol.

Reviewer 3 Report

I commend the authors on their manuscript entitled Reliability and validity of the ground reaction force asymmetric index at seat-off as a measure of lower limb functional muscle strength: a preliminary stud[y]. After review of the manuscript, I recommend minor editorial revisions. Since only minor editorial revisions are suggested I do not recommend individual line item responses by the authors, but rather thoroughly review and implement these suggestions where the authors see fit.

Editorial comments:

Title: Revise to “preliminary stud[y]”

Featured Application:

Line 15: Consider revising was measured as high to had high reliability

Line 17: and a period after test protocol.

Abstract:

lines 20-21: uncapitalize (1) designed, (2) cross-sectional, (3) general.

lines 20-21: This study was a cross-sectional design in a general hospital setting.

line 22: remove ‘was’

Introduction:

The set-up and/or justification for the new GRF asymmetry could use some minor revision in the introduction. Below are a few suggestions to consider.

Lines 45-46. Consider the following revision. While the manual muscle test (MMT) measures muscle strength, the subjective nature of the test lacks reproducibility. Consider removing  the poliomyelitis information and reference. After further review, I could consider removing this portion referencing MMT as it seems out of place considering it was not a measure assessed.

Line 51: after isokinetic dynamometry, [measures] present challenges and remove ‘among others’.

Lines 57-58. ‘are widely used to assess muscle strength’

Line 66. Consider expanding on what is asymmetry index and how that will assist with the proposed measure. In other words, connect how this is important to address limb differences.

Lines 66-67. Measurement methods (e.g., HHD, hand grip strength, STS).

Methods:

Line 135: expand on the actual cost of the force plates.

Line 179: remove period before reference 24.

Results:

Table 1. For height and weight add (units) next to each outcome.

Table 2. Add units next to each outcome (e.g., neutral STs – Walking Speed).

Tables 4 & 5. Consider reporting all p values next to each Pearson r value.

Figure 4. Report correlation values for A and B.

Figures 3 & 4. Visually, I do not know if these figures add to the manuscript. This information may be better presented in the text itself rather than in the current figure designs.

Discussion:

Lines 400, 421: No [significant] correlation.

Author Response

Response to Reviewer 3 Comments

We sincerely thank the reviewers for their constructive and valuable comments. We have thoroughly addressed their specific concerns and have introduced the suggested changes that were very helpful in improving the manuscript. The main changes in the revised manuscript are the red font color. We used the “Track Changes” function in Microsoft Word. Point-by-point responses to the reviewers’ comments are provided below.

I commend the authors on their manuscript entitled Reliability and validity of the ground reaction force asymmetric index at seat-off as a measure of lower limb functional muscle strength: a preliminary stud[y]. After review of the manuscript, I recommend minor editorial revisions. Since only minor editorial revisions are suggested I do not recommend individual line item responses by the authors, but rather thoroughly review and implement these suggestions where the authors see fit.

Editorial comments:

Title: Revise to “preliminary stud[y]”

We modified as followed > preliminary study

Featured Application:

Line 15: Consider revising was measured as high to had high reliability

  • We modified as followed.

Line 15-16 : The reliability of this new developed test measuring lower extremity muscle strength using force plate during sit-to stand performance was measured as high to had high reliability, but the validity was poor.

Line 17: and a period after test protocol.

  • We modified as followed.

Line 17: To be clinically useful, this test should be further refined by modifying the test protocol and method.

Abstract:

lines 20-21: uncapitalize (1) designed, (2) cross-sectional, (3) general.

  • We modified as followed.

Line 20-21 : designed as cross-sectional study in general hospital setting.

lines 20-21: This study was a cross-sectional design in a general hospital setting.

  • We modified as followed.

Line 20-21 : This study was a cross-sectional design in general hospital setting.

line 22: remove ‘was’

  • We modified as followed.

Line 22 : GRF AI assessed

Introduction:

The set-up and/or justification for the new GRF asymmetry could use some minor revision in the introduction. Below are a few suggestions to consider.

Lines 45-46. Consider the following revision. While the manual muscle test (MMT) measures muscle strength, the subjective nature of the test lacks reproducibility. Consider removing the poliomyelitis information and reference. After further review, I could consider removing this portion referencing MMT as it seems out of place considering it was not a measure assessed.

  • We modified as followed.

Line 48-49 : While MMT measures muscle strength, the subjective nature of the test lacks reproducibility.

Line 51: after isokinetic dynamometry, [measures] present challenges and remove ‘among others’.

  • We modified as followed.

Line 53 : In addition to MMT, conventional methods of measuring muscle strength include a hand-held dynamometer (HHD) and isokinetic dynamometry.

Lines 57-58. ‘are widely used to assess muscle strength’

  • We modified as followed.

Line 60 : In clinical practice, sit-to-stand (STS) and heel-raise tests are widely used to assess muscle strength.

Line 66. Consider expanding on what is asymmetry index and how that will assist with the proposed measure. In other words, connect how this is important to address limb differences.

  • We modified as followed.

Line 68-72 : We hypothesize that the GRF asymmetry index (AI) measurement is useful as a method for evaluating lower extremity strength. To verify the validity of the new test, the correlation with the conventional test was analyzed. Therefore, this study aimed to evaluate (1) the reliability of the new GRF AI at seat-off off and (2) the validity of this new test by comparing it with other muscle strength measurement methods (HHD and hand grip strength) and walking speed.

Lines 66-67. Measurement methods (e.g., HHD, hand grip strength, STS).

  • We modified as followed.

Line 71-72 : (2) the validity of this new test by comparing it with other muscle strength measurement methods (HHD and hand grip strength) and walking speed.

Methods:

Line 135: expand on the actual cost of the force plates.

  • We modified as followed.

Line 148 : To measure vertical GRFs, three low-cost force plates (Vernier Software & Technology, OR, USA), about 600 dollars each, were used.

Line 179: remove period before reference 24.

  • We removed period before reference 24.

Results:

Table 1. For height and weight add (units) next to each outcome.

  • We modified as followed.

At Table 1, we add units to each outcome.

Table 2. Add units next to each outcome (e.g., neutral STs – Walking Speed).

  • We modified as followed.

At Table 2, we add units to each outcome.

Tables 4 & 5. Consider reporting all p values next to each Pearson r value.

  • We modified as followed.

At Table 4 & 5, we add p values next to each Pearson r value.

Figure 4. Report correlation values for A and B.

Figures 3 & 4. Visually, I do not know if these figures add to the manuscript. This information may be better presented in the text itself rather than in the current figure designs.

  • We removed Figure 3, 4 in response to review comments

Discussion:

Lines 400, 421: No [significant] correlation.

  • We modified as followed.

Line 414, 435 : No significant correlation

Round 2

Reviewer 1 Report

The revised version of the manuscript has improved. All comments were sufficiently addressed.